# Understanding the Potential for Pharmacy Expertise in Palliative Care: The Value of Stakeholder Engagement in a Theoretically Driven Mapping Process for Research

**DOI:** 10.3390/pharmacy9040192

**Published:** 2021-11-26

**Authors:** Joseph Elyan, Sally-Anne Francis, Sarah Yardley

**Affiliations:** 1Foundation Programme, Barking, Havering and Redbridge University Hospitals NHS Trust, Queen’s Hospital, Rom Valley Way, Romford RM7 0AG, UK; joseph.elyan@nhs.net; 2Marie Curie Palliative Care Research Department, University College London, London W1T 7NF, UK; sally-anne.francis@ucl.ac.uk; 3Central & North West London NHS Foundation Trust, 350 Euston Road, Regent’s Place, London NW1 3AX, UK

**Keywords:** medication management, palliative care, end-of-life care, patient safety, stakeholder engagement

## Abstract

Potentially avoidable medication-related harm is an inherent risk in palliative care; medication management accounts for approximately 20% of reported serious incidents in England and Wales. Despite their expertise benefiting patient care, the routine contribution of pharmacists in addressing medication management failures is overlooked. Internationally, specialist pharmacist support for palliative care services remains under-resourced. By understanding experienced practices (‘what happens in the real world’) in palliative care medication management, compared with intended processes (‘what happens on paper’), patient safety issues can be identified and addressed. This commentary demonstrates the value of stakeholder engagement and consultation work carried out to inform a scoping review and empirical study. Our overall goal is to improve medication safety in palliative care. Informal conversations were undertaken with carers and various specialist and non-specialist professionals, including pharmacists. Themes were mapped to five steps: decision-making, prescribing, monitoring and supply, use (administration), and stopping and disposal. A visual representation of stakeholders’ understanding of intended medicines processes was produced. This work has implications for our own and others’ research by highlighting where pharmacy expertise could have a significant additional impact. Evidence is needed to support best practice and implementation, particularly with regard to supporting carers in monitoring and accessing medication, and communication between health professionals across settings.

## 1. Introduction

A global commitment and action to reduce avoidable medication-related harm for all patients has been sought since 2017, with a focus on high risk situations, polypharmacy, and transitions of care [1]. Prescribing errors alone affect 7% of medication orders, 2% of patient days, and 50% of hospital admissions [2]. In many palliative care systems, risks in medicines management are compounded by frequent use of opioids and sedatives [3], non-specialist care [4], movement between different care settings, and an integral reliance on carers (family, friends, neighbours) in medication-related activities [5,6]. Evidence investigating medicines management in palliative care is scarce and has failed to recognise the differences between intended and experienced processes. In the UK, medication management is the second most common reported serious incident (after pressure ulcers) for patients receiving palliative care [7]. Despite this, the potential for pharmacy expertise to address this issue is overlooked [8,9,10,11,12,13].

Prescribing for symptom control is just one step in a multi-step, socially constructed process of medicines management, in which a variety of professionals, carers, and patients must interact across, and within, different healthcare and home settings. As such, there are myriad factors that can disturb the intended processes and hence, safety [14]. It is recommended that pharmacists are included in the multi-disciplinary team involved with prescribing in palliative care [15,16], and that specialist palliative care settings integrate pharmacists with specialist knowledge [17,18]. Literature has shown that pharmacist involvement can improve patient care and reduce medication errors [19,20,21,22,23,24,25]. Nonetheless, there is limited commissioning of direct pharmacy support in palliative care, leaving other professionals to manage complex medication regimes and adjustments. 

We know that historically established divisions of labour and normative rules (practice etiquette) shape prescribing and medication use [26]. We are carrying out a study, ‘Getting Prescription Medications Right at Home, in Hospital & Hospice’ (Marie Curie Research Grants Scheme 2018, MC-19-904), that employs activity theory [27] in order to compare the intended processes with the experienced practices. We will be using patients’ activity systems as the unit of analysis, within and across the contexts of home, hospital, and hospice, to investigate and model practices of prescribing and medication use for symptom control. By generating an understanding of experienced practices (‘what happens in the real world’) in prescribing and medication use in palliative care, as distinct from the intended processes (‘what happens on paper’), patient safety issues can be more effectively identified and addressed. This commentary documents the first step in our research study; an engagement and consultation exercise to understand the direct experiences and concerns of a variety of stakeholders in palliative care. In doing so, we demonstrate the effectiveness of investing in high-quality stakeholder engagement and encourage others to employ it in their own work. Our research will be directed by essential issues raised through the stakeholder work, and so ensuring comprehensiveness and attentiveness to what matters most, based on carer and professional stakeholders’ perspectives. We hope that this commentary will help others use similar engagement methods to prioritise areas for further research and improvement, as well as stimulate wider debate on the contribution of pharmacists within multi-disciplinary palliative care teams.

## 2. Stakeholder Engagement

Our future empirical research will gather the experiences of patients, carers, and professionals across home, hospital, and hospice through observations and interviews. Prior to this, to develop a model of the intended multi-step process, we are conducting a scoping review [28,29,30]. It is recommended that scoping reviews include a ‘consultation’ step with various stakeholders to provide additional information, perspectives, meaning, and applicability [30]. It is argued that the consultation component of scoping studies provides methodological benefits and increased effectiveness in communicating findings to relevant stakeholders [28]. Despite this, best practice for carrying out stakeholder engagement remains unclear [28]. 

In line with recommendations for a consultation step, informal conversations were undertaken with carers who had experience of, or an interest in, managing palliative medications for others, and a variety of specialist and non-specialist healthcare professionals, including pharmacists (Table 1), to help shape our model of intended processes and direct our evidence scoping work. Stakeholder participation was voluntary; people contacted us following the circulation of adverts on social media, patient and public involvement recruitment websites, and through our own and our study funder’s networks. We hoped that it would provide a multi-voiced perspective on what to focus on and prioritise, in both our scoping review and empirical study. We believe that our active engagement of relevant stakeholders will also help enable effective dissemination and translation into practice and policy. 

The conversations were guided by five steps we considered potentially critical to the process of medicines management in palliative care: decision-making, prescribing, monitoring and supply, use (administration), and stopping and disposal. We allowed stakeholders to lead the conversation based on their experience and expertise, rather than asking direct questions about who should do what. This was deliberate, as we wanted to understand their priorities without priming them to particular conclusions. Common themes were identified through open mind-mapping and discussion (led by the first author, all authors discussed and refined) before being mapped to the five suggested steps. Additional areas of interest were also documented, and a visual model of intended medicines processes was produced, highlighting areas of relevance for our scoping review and later empirical work. The monitoring and supply of medications was the most prominent theme. Consequently, it has become a priority for our research, and it is an area where pharmacy expertise could have a significant impact.

## 3. Themes

### 3.1. The Decision to Prescribe

Prescription decision-making was predominately discussed by professional stakeholders. A major theme identified was the balancing of protocols and expertise; professionals’ views varied on whether they relied on protocols or what several stakeholders referred to as ‘gut feeling’, when making decisions concerning medication. This depended on the stakeholders’ exposure to palliative medicines; specialist palliative care professionals were more comfortable drawing upon expertise to inform prescribing decisions. They felt able to safely deviate from protocols to achieve more personalised, patient-centred care. This is supported by evidence that suggests that specialist palliative care professionals regularly use experience to prescribe off-label medications [31]. Comparatively, general practitioners (GPs) felt under-confident in prescribing palliative medicines without protocols. They wanted protocols to remind them of correct dosages and regimes, rarely straying from these. This is reflected in evidence that suggests that non-specialist prescribers are cautious with their approach to palliative medicines due to fear of hastening death [32], and that effective anticipatory prescribing is heavily dependent on a prescriber’s level of experience [33]. Despite their perceived importance, professional stakeholders described frustrations with either inconsistent or non-existent protocols, particularly with non-opioid medications. 

Both professionals and carers highlighted the importance of communicating decisions to the patient and carers. This was a particular point of emphasis for the pharmacists who felt that it was essential to communicate not only what the medications are, but also their purpose, and specific side effects that carers could monitor. This was particularly pertinent in the community, as carers were able to recognise medication issues more easily than professionals due to time spent with the patient. Carers discussed how effective communication regarding prescribing decisions made the entirety of the process easier to navigate. This finding corroborates with evidence that found that the effective communication of decisions was one of the most important aspects of palliative care for patients [34]. Professionals identified the significance of understanding a carer’s and patient’s level of knowledge, allowing them to adjust their communication to suit the patient’s needs. Despite acknowledging the importance of communication, carers often felt that it varied in practice, leaving them to work things out for themselves. Stakeholders suggested multiple reasons for this, of which some are supported in the evidence, such as patients and carers being overloaded with information [35], emotional distress affecting retention of information [36], and time or resources limiting professionals’ capacity.

### 3.2. The Act of Prescribing

Stakeholders explained that they were often unsure of how quickly to increase doses of medications, and that it was difficult to balance symptom control with over-medicating. This was made more challenging by a lack of guidance and evidence to support decisions, which is supported in recent research [37,38,39]. As with decision-making, experience played a central role in a prescriber’s confidence in titrating medications. Many of the specialist palliative care prescribers and pharmacists spoke of ‘tinkering’ with medications, described by one pharmacist as carrying out ‘safe-to-fail’ experiments by prescribing, monitoring effects and adjusting accordingly. They felt that this was fundamental to effective, patient-centred symptom control. Tinkering was scarcely discussed by non-specialist prescribers, potentially exposing a lack of confidence in adjusting medications and doses. Evidence supports the notion that monitoring and adjustment is heavily dependent on a prescriber’s level of experience [32]. 

Stakeholders considered prescribing in the community to present unique challenges. Anticipatory prescribing has received recent research interest [33,40,41,42,43,44,45,46,47], and its significance was reflected in our conversations. GPs stressed the importance of effective anticipatory prescribing, as it allows patients, carers, and professionals to manage symptoms out-of-hours without delay. Carers and pharmacists agreed that anticipatory medications reduce anxieties about medication access if a patient’s condition deteriorates. Nevertheless, stakeholders recognised the difficulties and risks of anticipatory prescribing. Professionals felt that knowing when to prescribe was challenging; it required a balance between preparedness and the risk of abuse (e.g. diversion), waste, and burdening carers. Carers described how anticipatory medications can contribute to the overwhelming number of medications at home, and pharmacists discussed how the controlled palliative medicines can leave carers with significant responsibility.

A significant frustration identified by carers in the community was the lengthy process of accessing prescribers; often they must inform the district nurse, who then communicates the need to a prescriber, who must then prescribe and supply the medicines. This time-consuming process may leave the patient with uncontrolled symptoms, resulting in potential harm. These issues were heightened in out-of-hours situations where prescribers are scarce and are cautious to alter medication doses [48]. Poor direct access to prescribers led to carers feeling overwhelmed and stressed about not managing patients’ symptoms effectively. 

Deprescribing was an area that pharmacists felt was particularly poorly managed. Deprescribing is the process of discontinuing potentially harmful medications that have a minimal benefit to the improvement of a patient’s quality of life [49]. Several pharmacists and physicians felt that deprescribing a patient’s regular medications was so important that it should have its own step in the process. Stakeholders discussed how effective deprescribing can reduce the medication burden on patients and carers, reduce the risk of side effects, and avoid medication waste. These views are reflected in the limited evidence [49]. Yet, stakeholders highlighted difficulties in effectively deprescribing, including limited time and questions concerning responsibility. GPs felt that it was often unclear whether it was their responsibility to deprescribe, particularly when a patient had been discharged from an acute or specialist setting. Some assumed that if regular medications could be discontinued, then that would have occurred prior to discharge. Potentially challenges such as these are what leads to the ineffective deprescribing seen when people are receiving palliative care [50].

### 3.3. Monitoring and Supply of Medications

Primarily, carers discussed numerous challenges that disrupted the intended medicines process. Stakeholders often felt uncertain of whose responsibility it was to monitor the efficacy and side effects of medications. The prescriber, district nurses, carers, and patients were all suggested as responsible; however, this uncertainty resulted in carers feeling ultimately responsible. GPs often waited to be informed of issues, rather than actively monitoring patients due to time-pressures, with carers emphasising the importance of district nurses. However, due to the variability in the confidence of district nurses, the responsibility often fell to the carer to recognise side effects and act quickly and safely without training or professional support.

Access to medications in the community was recognised as a major disturbance to the intended process; a view that is reflected in the literature [51]. Various stakeholders, particularly carers, discussed difficulties with medication availability at pharmacies. They spoke of having to travel to multiple pharmacies to obtain the required medications, or of being denied access to a complete prescription if the pharmacy did not stock all of the items. Pharmacists recognised the difficulties of access and discussed how certain localities had designated pharmacies for palliative medicines, yet not all carers were made aware of this. Carers felt that it was their responsibility to seek out the availability of palliative medications from their local pharmacy. However, carers were often limited in their abilities to travel to pharmacies due to patient need, medical or mobility issues, subsequently relying on other family, friends or neighbours to collect medications. This was made significantly easier if pharmacies delivered medications. Most carers concluded that monitoring and supply was the biggest barrier to an intended medicines process; if the difficulties had been explained to them in advance, their preferred place of care for the patient may not have been at home.

### 3.4. Use (Administration)

Medication administration was spoken of predominately by carers, suggesting that it was a significant community concern. Responsibility arose again as a theme; it was unclear whose role it was to administer medications in the community. It was often the role of district nurses; however, their confidence and experience with dealing with palliative medications varied, corroborating with recent evidence [44]. Carers and patients were responsible for administering oral medications, contributing to the burden on carers. Carers’ views on administration varied; some, particularly those with healthcare backgrounds, were frustrated at being unable to administer injectable medications. Research suggests that carers who administer injectable medications feel a sense of achievement and empowerment [52]. When in hospital, carers discussed the delays between the requesting of medications and its administration by nursing staff, exemplified by asking the incorrect nurse, ward distractions, and the time taken between obtaining the medication and the patient receiving the dose.

Medication adherence is reportedly poor in palliative care patients [53,54,55,56]. Our work highlighted this as an area of concern, with pharmacists, in particular, providing an insight into the barriers faced by patients. Adherence was difficult for patients due to complex medication regimens, but this was aided by effective deprescribing. Palliative patients often experienced swallowing difficulties and were thus unable to take standard oral medications. This was worsened by the size and shape of medications. Carers discussed the packaging of medications; they noted that identical medications were packaged differently depending on the pharmacy, making it confusing for those administering them. This was compounded when patients were discharged from hospital with medications that looked, and were packaged, differently to their usual medicines.

### 3.5. Stopping and Disposal

Stopping and disposal of medication is scarcely discussed in the literature. Similarly, stakeholders rarely discussed it without being prompted. However, it was considered by most to be an important, and often overlooked, step in the process. It was exclusively discussed in relation to community palliative care. Generally, stakeholders recognised that medications should be returned to a pharmacy for disposal once they are no longer required. In the UK, pharmacies are the only legal entities allowed to receive medications for disposal, and named patient medications cannot currently be reused, although the COVID-19 pandemic has resulted in further calls for this to be changed [46]. Despite this, carers and pharmacists described incidences of palliative medications still being stored at home, highlighting the associated risks of abuse, including the diversion and wastage of medications. Stakeholders suggested several possible reasons contributing to this, such as ineffective checking of repeat prescriptions, duplicated supply of medications when a patient is discharged from hospital, and a lack of clear guidance on whose responsibility it is to follow up on medication disposal. Stakeholders recognised the juxtaposition between the stringent processes when a patient is prescribed controlled medications and the lack of follow-up when a patient has died. These are issues that should be addressed within healthcare systems and internationally different approaches might be usefully shared.

### 3.6. Contextual Issues

Throughout our engagement work, several contextual themes were identified that created disturbances across multiple steps. Communication between professionals and services was essential, but often ineffective. This was predominantly discussed in reference to the discharge of patients from hospital; GPs spoke of the poor communication of decisions and plans from hospitals. Communication was also important between GPs and district nurses so that patients’ symptoms and concerns could be effectively managed. This highlighted the potential role of the GP as a director by delegating and identifying the role of healthcare professionals in patients’ palliative care. Despite their pivotal role in community palliative care, pharmacists discussed the lack of communication between other healthcare professionals and themselves, resulting in a lack of clarity regarding a patient’s management. Interdisciplinary communication has been identified as pivotal in providing effective palliative care [47], yet we recognise that this is still poorly achieved. 

Continuity of care can enable more personalised care and provide patients with easy access to help [57], however, we found that this is not always achieved. Carers found it difficult to repeatedly communicate the dynamic nature of a patient’s symptoms, wishes, and condition to different professionals. They felt that this was difficult in all settings, but particularly out-of-hours in the community. Finally, pharmacists, nurses, and junior doctors found it difficult to question senior doctors’ decisions when prescribing palliative medicines, due to the hierarchical nature of healthcare [26], which could perpetuate poor practice.

### 3.7. Visual Intended Medicines Process

Following our engagement work, we created a visual representation of the components required for any intended medicines process, as identified and described by our stakeholders (Figure 1). We have mapped each point to one of the five steps discussed in this commentary. This highlights how the process can be superficially linear, but in fact, involves several feedback loops. We will seek evidence on, if, and/or how, these loops are integrated into intended systems in our scoping review. We wanted to gain insight into the three settings of hospital, home, and hospice, and the transition between these. However, as hospice was scarcely discussed in our engagement work it will be a point of emphasis in our further empirical work.

## 4. Concluding Remarks

### 4.1. Implications for Our Own and Others’ Research

This engagement and consultation exercise has enabled us to gather direct experiences from a variety of stakeholders, demonstrating the value of this early step in underpinning our scoping review and empirical work. We have identified key themes and produced a visual representation of the components required in any intended palliative medicines process (Figure 1). As the most spoken about theme, monitoring and supply of medications will be a focus of our work going forward. Responsibility was another significant theme; it was clear that there was uncertainty regarding responsibility for numerous steps in the intended processes, resulting in carers feeling ultimately responsible. It is essential that our work, and future research, investigates this further and identifies potential strategies to reduce the burden on carers. Our engagement work was limited by only speaking to one nursing stakeholder, and the wider research in this area is also limited [58,59]. With district nurses playing a pivotal role in community palliative care, we suggest that this too should be a focus of future research.

### 4.2. Implications for Pharmacists and Commissioners of Pharmacy Services

Involving pharmacists as a core member of the palliative care multi-disciplinary team is not common, but with correct provisions this could prove pivotal in improving palliative medicines processes across all steps, from the decision to prescribe through to stopping and disposal [60]. Future research should address whether it is realistic for pharmacists to meaningfully assist at scale within healthcare systems. As well as identifying areas for future research, we have identified several areas where all pharmacists have expertise that could play an essential role. These align with a recent report from the Royal Pharmaceutical Society Wales into the pharmacy’s contribution to palliative care [61]. Our pharmacist stakeholders agreed that specialist palliative care pharmacists could be an effective collaborator with non-specialist prescribers across all settings [62], either through an independent prescribing role (e.g., as non-medical prescribers—in the UK this term includes all prescribers who are not doctors), the development of effective guidelines and protocols, or by providing patient-specific advice. In the community, pharmacists are essential in supporting carers and could help combat the issues raised by our carer stakeholders by identifying and managing side effects, providing access to prescribing expertise, improving the monitoring and supply of medications within the system, communicating medication availability, and delivering medications to patients. The disposal of medications was an area that we identified with significant disparity between intended and actual processes; our stakeholders felt that community pharmacists could prove crucial by taking ownership of disposal and ensuring the timely retrieval of medications from a patient’s home. Professional stakeholders emphasised how all pharmacists have expertise that should be integrated into the communication between professionals and patients. Pharmacists could provide verbal or written communication to patients to troubleshoot queries, debunk myths, and translate prescriptions into practical advice for patients and carers [62]. This could be particularly effective when patients are transitioning between different settings. Finally, pharmacists were rarely mentioned when discussing communication between services. This ultimately highlights the need for the expertise of pharmacists to be actively integrated into the medicines management process, and for them to feel empowered enough to involve themselves in palliative care.

## Figures and Tables

**Figure 1 pharmacy-09-00192-f001:**
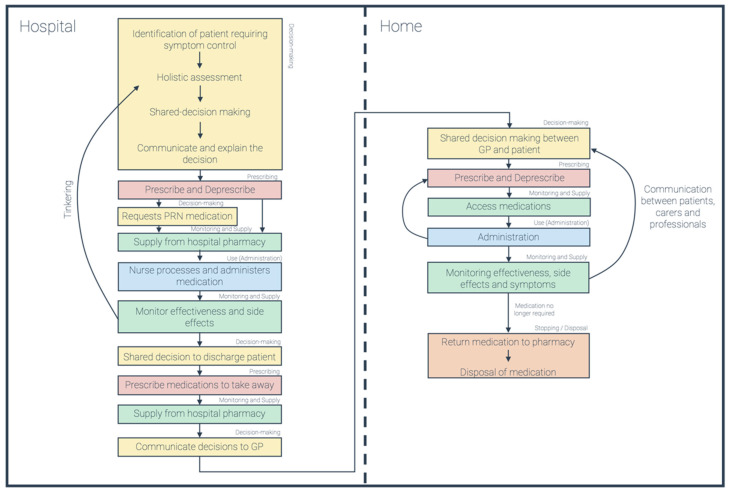
Intended Medicines Process. Each colour represents a step discussed in this piece: decision-making (yellow), prescribing (red), monitoring and supply (green), administration (blue), and stopping and disposal (orange).

**Table 1 pharmacy-09-00192-t001:** Participant stakeholders: informal conversation participation by videocall, *n* = 20, participation by email discussion, *n* = 1.

Stakeholder	Background	Total
1	Non-clinical researcher	3
2	Non-clinical researcher	
3	Non-clinical researcher	
4	General practitioner	3
5	General practitioner	
6	General practitioner (with specialist palliative care interest)	
7	carer	7
8	carer	
9	carer	
10	carer	
11	carer	
12	carer	
13	carer	
14	Pharmacist (non-specialist)	4
15	Pharmacist (non-specialist)	
16	Pharmacist (non-specialist)	
17	Pharmacist (specialist)	
18	Specialist palliative care professional (nurse)	4
19	Specialist palliative care professional (physician)	
20	Specialist palliative care professional (physician)	
21	Specialist palliative care professional (physician)	

## Data Availability

No new research data were created or analyzed in this work. Stakeholders were not asked to agree to data sharing as this work was conducted using patient and public engagement principles rather than within a research framework.

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
