# Peer review of "Understanding the Potential for Pharmacy Expertise in Palliative Care: The Value of Stakeholder Engagement in a Theoretically Driven Mapping Process for Research"

_pharmacy, 2021, doi:10.3390/pharmacy9040192_

Round 1

Reviewer 1 Report

Thank you for your submission.

There is very little detail on the qualitative methodology. How were the participants selected? Were these one-on-one interviews? How did coding and theming of the data occur? How many people conducted this independently?

Did the pharmacists have views on prescribing? Could this task be delegated to trained pharmacists?

Given the title of the manuscript, were the various stakeholders actually asked specifically about the current and potential role of pharmacists in palliative care? This should be clear in the manuscript.

The involvement of GPs in the management of the patient and medication availability seem to be particularly poor. Is this generally the case across the UK? It sounds very suboptimal for a highly-developed Western country. Are there any dedicated community-based palliative care services?

On several occasions, the authors refer to the associated risks of abuse and wastage of medications. The risk of diversion is probably more important.

The discussion concludes with the potential role of pharmacists, which seems largely ignored by other stakeholders. This is not unusual in the literature. Is it therefore realistic that pharmacists can meaningfully assist on a large scale? Why isn’t it happening already? Lack of time? Lack of skills? Furthermore, if they themselves didn’t discuss pharmacist prescribing, what genuine commitment and interest is there?

The authors shouldn’t self-assess their research as “This high-quality stakeholder engagement exercise..”.

Reviewer 2 Report

Manuscript No. 1472913

„Understanding the potential for pharmacy expertise in palliative care: the value of stakeholder engagement in a theoretically driven mapping process for research” by Joseph Elyan, Sally-Anne Francis and Sarah Yardley for Pharmacy

Comments:

The work is interesting, showing many problems resulting from both the technical aspects of drug circulation as well as factors resulting from interpersonal communication and responsibility related to palliative treatment.

  1. Paragraph 3.5. Disposing of drugs is a difficult topic. However, is it not worth considering briefly in the text whether only pharmacies should be the point where unused drugs are returned. If they are not overdue, could hospices, of course, after medical consultations, not be recipients of such medications? Please, very briefly supplement this chapter with various possibilities of drug circulation that can no longer be used by a particular patient and can be used by others.

Round 2

Reviewer 1 Report

Thank you for the revision.